# Risk of consecutive esotropia after surgery for intermittent exotropia according to passive duction force

**Hyunkyoo Kang[1], Hyun Jin Shin [2]\*, Andrew G. Lee[3,4,5,6,7,8,9,10]**

**1** Department of Mechatronics Engineering, Konkuk University Glocal Campus, Chungcheongbuk-do, Republic of Korea, **2** Department of Ophthalmology, Research Institute of Medical Science, Konkuk University Medical Center, Konkuk University School of Medicine, Seoul, Republic of Korea, **3** Department of Ophthalmology, Blanton Eye Institute, Houston Methodist Hospital, Houston, TX, United States of America, **4** Department of Ophthalmology, Neurology, Neurosurgery, Weill Cornell Medicine, New York, NY, United States of America, **5** Department of Ophthalmology, University of Texas Medical Branch, Galveston, TX, United States of America, **6** Department of Ophthalmology, UT MD Anderson Cancer Center, Houston, TX, United States of America, **7** Department of Ophthalmology, Texas A and M College of Medicine, College Station, TX, United States of America, **8** Department of Ophthalmology, University of Iowa Hospitals and Clinics, Iowa City, IA, United States of America, **9** Department of Ophthalmology, Baylor College of Medicine and the Center for Space Medicine, Houston, TX, United States of America, **10** Department of Ophthalmology, University of Buffalo, Buffalo, NY, United States of America

\* shineye@kuh.ac.kr

## Abstract

### Purpose

To determine the relationship between consecutive esotropia (ET) and passive duction force (PDF) in patients with intermittent exotropia (XT).

### Methods

The study enrolled 70 patients in whom PDF was measured under general anesthesia prior to XT surgery. The preferred eye for fixation (PE) and the nonpreferred eye for fixation (NPE) were determined using a cover–uncover test. The patients were subdivided into two groups according to the angle of deviation at 1 month postoperation: (1) consecutive ET (CET group), >10 prism diopters (PD) of ET; and (2) non-CET (NCET group), $\leq$10 ET or residual exodeviation. The relative PDF of the medial rectus muscle (MRM) was obtained by subtracting the ipsilateral PDF of the lateral rectus muscle (LRM) from the PDF of the MRM.

### Results

The PDFs for the LRM in the PE in the CET and NCET groups were 47.28 g and 58.59 g, respectively ($p = 0.147$), and 56.18 g and 46.59 g for the MRM ($p = 0.11$), and in the NPE were 59.84 g and 55.25 g, respectively, for the LRM ($p = 0.993$), and 49.12 g and 50.53 g, respectively, for the MRM ($p = 0.81$). However, in the PE, the PDF in the MRM was larger in the CET group than in the NCET group ($p = 0.045$), which was positively associated with the postoperatively overcorrected angle of deviation ($p = 0.017$).

**Data Availability Statement:** All relevant data are within the manuscript and its Supporting Information files.

**Funding:** This research was supported by Basic Science Research Program through the National

Research Foundation of Korea(NRF) funded by the
Ministry of Education (2020R1I1A3075301). This
sponsor had no role in the design or conduct of
this research.

**Competing interests:** The authors have declared
that no competing interests exist.

## Conclusions

An increased relative PDF in the MRM in the PE was a risk factor for consecutive ET after
XT surgery. Quantitative evaluation of the PDF could be considered when planning strabis-
mus surgery to achieve the desired surgical outcome.

## Introduction

Intermittent exotropia (XT) is the most common form of strabismus, with surgery being the
most common treatment [1]. It has been widely accepted that the initial postoperative align-
ment performed for XT should target esodeviation to ensure favorable long-term motor align-
ment, due to a tendency of postoperative drifting toward XT [2–4]. Oh and Hwang [5]
concluded that 1-day postoperative overcorrection was the only factor that guarantees a suc-
cessful long-term outcome after XT surgery. Lee and Lee [6] also suggested that an overcorrec-
tion of 1–10 prism diopters (PD) following unilateral lateral rectus muscle (LRM) recession—
medial rectus muscle (MRM) resection (R&R) can produce good results.

However, even though most patients with an initial overcorrection after XT surgery exhibit
a drift to XT, a variable degree of esotropia (ET) can persist. The incidence of consecutive ET
after surgery for XT has been reported as 6–20% [7, 8]. Persistent consecutive ET can cause
undesirable outcomes such as diplopia and visual confusion [9, 10], especially in children,
where consecutive ET can lead to worsened stereopsis and amblyopia [11].

The usefulness of evaluating the mechanical properties of the extraocular muscles (EOMs)
by measuring their passive duction forces (PDFs) has been reported when planning XT sur-
gery due to the structural remodeling of EOMs in strabismus [12–17]. We previously devel-
oped a novel device to quantitatively measure PDF in EOMs [18], and have found the device
to be valid, reproducible, and reliable for measurements in normal subjects and patients with
XT [19, 20].

The aim of this study was to determine the relationship between consecutive ET and the
PDF in the LRM and MRM of patients with XT and to determine whether measuring the PDF
of EOMs can reduce the incidence of consecutive ET after XT surgery.

## Materials and methods

This prospective study was conducted at the Department of Ophthalmology of Konkuk Uni-
versity Medical Center in Seoul, Republic of Korea between January 2019 and April 2022. It
was approved by the Institutional Review Board and Ethics Committee of Konkuk University
Medical Center (registration number: KUH1100071). The study was conducted according to
the principles of the Declaration of Helsinki, with informed consent obtained from all
included participants and/or their parents/caregivers.

### Participants

This study enrolled 70 patients who were scheduled to receive unilateral R&R (nonpreferred
eye for fixation [NPE]) under general anesthesia to correct XT. All of the patients had a basic
XT pattern with a score on the objective control scale of the Newcastle Control Score of 1 or 2
[21] and did not have (1) dissociated vertical deviation, (2) >4 PD of vertical deviation, (3) lat-
eral incomitance, (4) an A or V pattern strabismus, (5) significant superior oblique or inferior
oblique under-/overaction, or (6) simulated divergence excess/convergence insufficiency.

The following exclusion criteria were also applied: (1) history of other ocular conditions (e.g., nystagmus, ptosis, or orbital diseases), (2) anisometropia or amblyopia, (3) eye movement limitation on duction/version tests, (4) thyroid disorder or muscular or neurological diseases (e.g., cerebral palsy or myasthenia gravis), (5) history of ocular trauma or previous ocular or periocular surgery, (6) history of receiving medications known to affect muscle tension (e.g., muscle relaxants) within the past month, (7) mean spherical error >+4.0 or <−6.0 diopters, or (8) developmental delay. Lateral incomitance was defined as a change in lateral gaze of >5 PD from the primary position. Anisometropia was defined as a spherical or cylindrical difference of >1.5 PD between the two eyes. Amblyopia was defined as a difference in the best-corrected visual acuity between the eyes of more than two Snellen lines (logMAR = 0.2).

## Passive duction force measurement

We used a previously described tension measuring device to quantitatively and continuously measure the PDF in EOMs (Fig 1). PDF measurements of each horizontal rectus muscle in both eyes were made under general anesthesia before XT surgery, as described previously [19, 20]. Anesthesia was induced by administering 5 mg/kg of sodium thiopental. A rocuronium (Esmeron®, MSD, Seoul, Korea) dose of 0.6 mg/kg was administered for muscle relaxation under the guidance of peripheral neuromuscular transmission monitoring (TOF-Watch SX®, Organon, Dublin, Ireland). The maximum PDF in each rectus muscle was recorded and analyzed. The measurements were performed by a single examiner (H.J.S.).

## Outcome measurements

Patients with diplopia associated with postoperative ET were managed using full-time monocular (fellow eye) patching for 1–4 weeks until the diplopia was resolved [22]. Consecutive ET

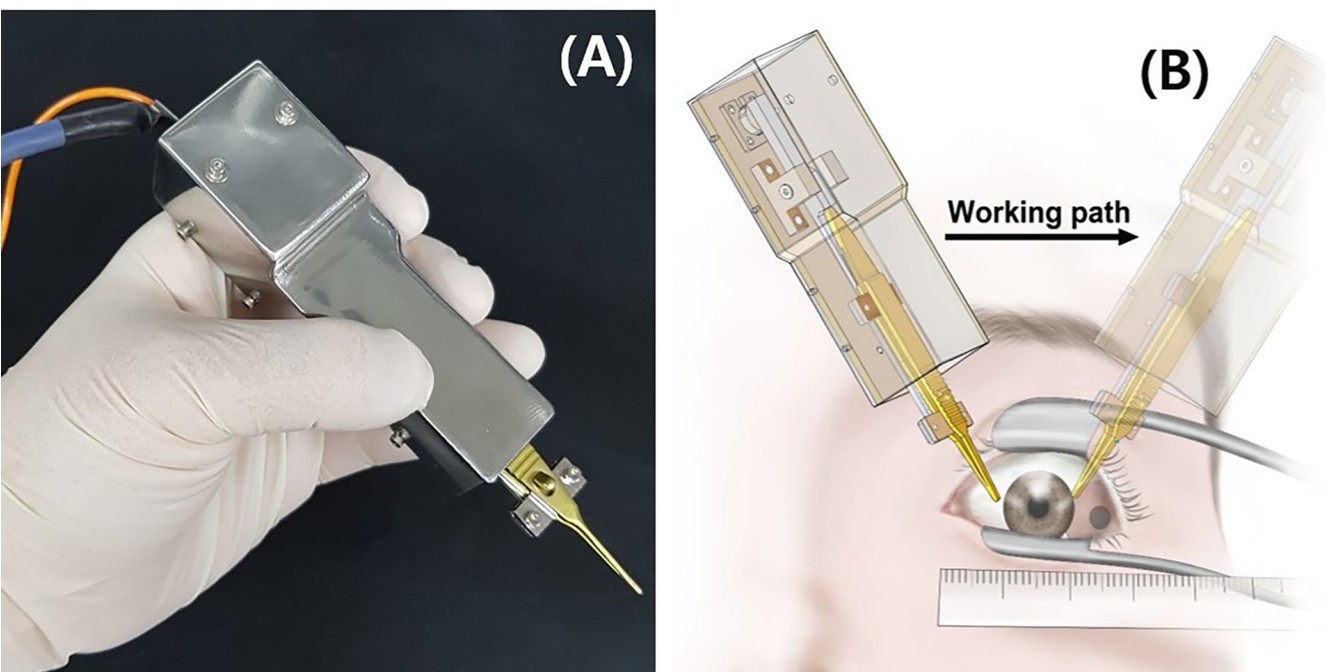

**Fig 1.** **(A)** Photograph of the device for measuring the passive duction force (PDF). **(B)** Schematic of measuring the PDF (reprinted with permission [20]). Locking forceps were attached to the limbus and rotated away from the direction of force to be tested (opposite side of muscle being tested). The operation method was similar to that of conventional forced duction tests with forceps.

was defined as the mean angle of the near and distance deviation >10 (PD) ET at 1 month postoperation, as described previously [9]. The patients were subdivided into two groups according to the presence of consecutive ET: consecutive ET (CET group) and non-CE (NCET group). The main outcome measures were the PDF in the LRM and MRM, and the relative PDF in the MRM, which was obtained by subtracting the ipsilateral PDF of the LRM from the PDF of the MRM.

Patient characteristics that were compared between the two groups included sex, age at surgery, refractive error, duration of manifest intermittent eye misalignment, dominant eye, stereoacuity according to the Titmus Stereoacuity Test, and angles of deviation at distance (6 m) and near (0.33 m). In the dominance test, the preferred eye for fixation (PE) was determined using repeated examinations of the cover–uncover test as well as from parental/caregiver and patient reports of the more frequently nondeviating eye [12].

## Statistical analyses

All calculations and statistical analyses were performed using Minitab software (version 21.1.1, Minitab, State College, PA, USA). The Anderson-Darling test was used to determine whether the data conformed to a parametric (Gaussian) or nonparametric (non-Gaussian) distribution. Sex and PE differences were analyzed using the chi-square test. Age at surgery, refractive error, duration of manifest deviation, stereoacuity, angle of deviation, and the PDF were analyzed using the Mann-Whitney $U$ test. The two eyes of each patient were compared using paired-sample $t$-tests. The linearity of relationships between the angle of deviation and PDF was evaluated using the Pearson correlation coefficient. The data are presented as mean±standard-deviation values, and the criterion for significance was set as $p<0.05$.

## Results

This study analyzed the 140 eyes of 70 patients with XT (age, 10.83±9.39 years). Anderson-Darling tests indicated that the PDFs of NPE and PE conformed to Gaussian and non-Gaussian distributions, respectively. The average angle of the near and distance deviations was 27.31±9.45 PD among all of the eyes. Consecutive ET occurred in 8 of the 70 patients (11.4%). The baseline characteristics (e.g., sex, age at surgery, refractive error, angle of deviation, and duration of manifest deviation) did not differ significantly between the CET and NCET groups (Table 1).

**Table 1. Clinical characteristics of study subjects in total groups.**

| Clinical Variables | CET (n = 8) | NCET (n = 62) | P value |
|---|---|---|---|
| Sex (male/female) | 5 / 3 | 29 / 33 | 0.402 [a] |
| Age at operation (years) | 10.25±2.96 | 10.9±9.94 | 0.150 [b] |
| Refractive error (SE)* | -1.31±1.21 | -1.42±1.67 | 0.313 [b] |
| Duration of manifest deviation (years) | 3.94±2.31 | 3.5±4.89 | 0.121 [b] |
| Stereopsis | 1.80 | 1.97 | 0.115 [b] |
| Angle of near and distance deviation (PD) | 24.47±9.02 | 27.68±9.52 | 0.150 [b] |
| Distance | 21.13 | 24.07 | 0.459 |
| Near | 27.80 | 31.29 | 0.321 |
| Mean of near and distance deviation | 24.47 | 27.68 | 0.370 |

SE, spherical equivalent

*, mean SE of both eye.

[a] p value relate to Chi-Square test

[b] p value relate to Independent-samples t-test.

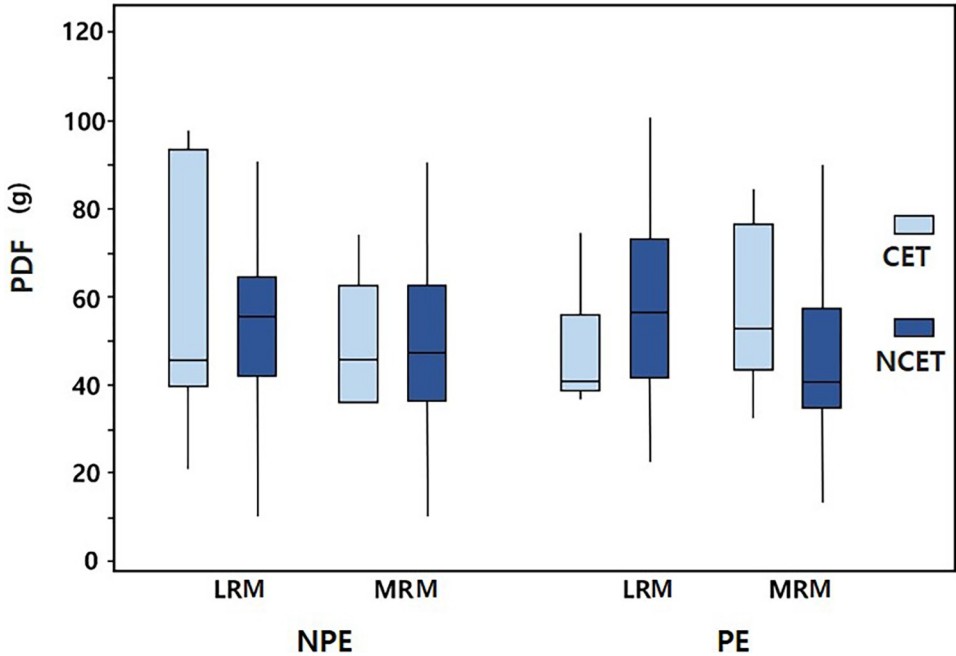

**Fig 2. PDFs of the horizontal rectus muscle in the preferred eye for fixation (PE) and the nonpreferred eye for fixation (NPE) in the consecutive esotropia (ET) group (CET group) and nonconsecutive ET group (NCET group).** In each box plot, the middle horizontal line indicates the median value, the boxes indicate the interquartile interval (25th and 75th percentiles), and the whiskers indicate the range. LRM, lateral rectus muscle; MRM, medial rectus muscle.

Fig 2 presents the measured PDFs of the horizontal rectus muscles in the PE and NPE in the two groups. There were no significant intergroup differences in the PDF between the two groups. The PDFs for the LRM in the CET and NCET groups were 47.28 g and 58.59 g, respectively ($p$ = 0.147), and 56.18 g and 46.59 g for the MRM ($p$ = 0.11) in the PE, and 59.84 g and 55.25 g, respectively, for the LRM ($p$ = 0.993), and 49.12 g and 50.53 g, respectively, for the MRM ($p$ = 0.81) in the NPE.

Fig 3 presents the measured relative PDF in the MRMs in the PE and NPE. In the NPE, there was no significant intergroup difference between the relative PDFs in the MRM (−10.72 ±26.28 g and −4.72±16.83 g, respectively; p = 0.549). However, in the PE, the relative PDF in the MRM of the CET group (8.90±23.43 g) was significantly larger than that of the NCET group (−12.0±17.1 g, $p$ = 0.045). The relative PDF in the MRM in the PE also had a positive association with the postoperative angle of esodeviation (Pearson correlation coefficient = −0.284, $p$ = 0.017) (Fig 4).

## Discussion

Consecutive ET is a persisting and often variable esodeviation that may occur following strabismus surgery to correct XT. The aim of the present study was to elucidate whether the PDF of EOMs are a predictive factor for consecutive ET after XT surgery. We evaluated the quantitative PDF in the horizontal rectus muscles (the LRM and MRM) of patients with XT and compared the PDFs between two groups (CET and NCET groups). The mean PDFs of the horizontal rectus muscles in the PE or NPE did not differ significantly between the two groups. The relative PDF in the MRM in the NPE did not different significantly between the two groups, but there was a significant difference in the relative PDF in the MRM in the PE. In

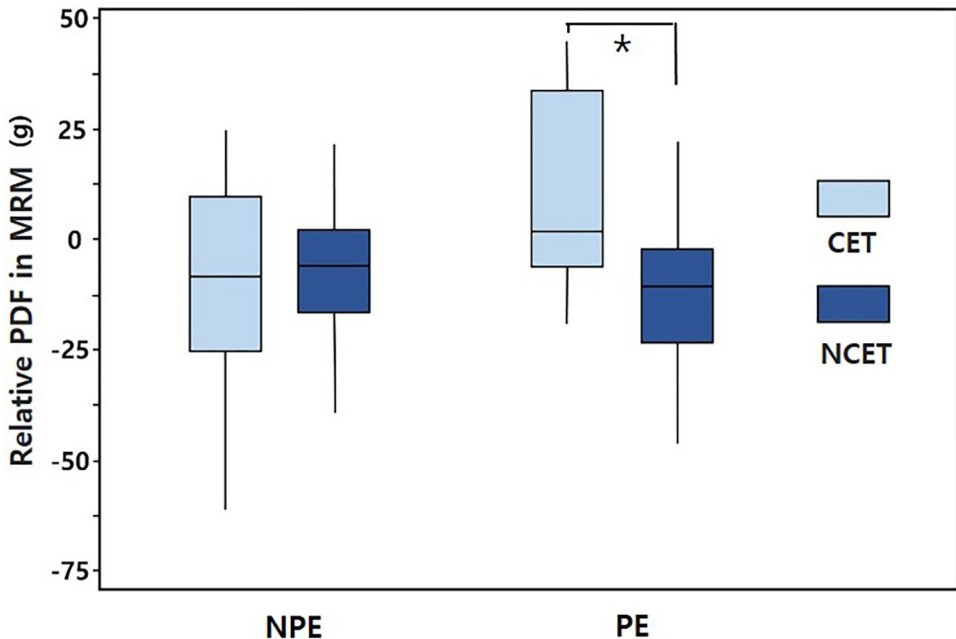

**Fig 3. Relative PDFs in the medial rectus muscle (MRM) in the PE and NPE in the CET and NCET groups.** In each box plot, the horizontal line indicates the median value, the boxes indicate the interquartile interval (25th and 75th percentiles), and the whiskers indicate the range. Asterisk indicates a statistically significant difference ($p = 0.042$).

addition, the relative PDF in the MRM in the PE increased with the amount of excessive over-correction after XT surgery.

A particularly interesting finding of the present study was that the risk of consecutive ET increased with the PDF of the MRM relative to the ipsilateral LRM in the PE, whereas there was no association between the PDF in the LRM in the NPE and consecutive ET. In our previous study, we demonstrated that the PDF of the LRM relative to the ipsilateral MRM in the NPE was larger than normal and could increase with the duration of manifest deviation and the angle of deviation in patients with XT. However, in the present study, the duration of manifest deviation and the angle of deviation did not differ between the CET and NCET groups. Kim et al. [14] also found that unilateral R&R surgery performed on the eye with more PDF in the LRM resulted in a more successful alignment and lower recurrence. We believe that increased PDF in the LRM in the NPE is not a predisposing risk factor for consecutive ET, although recession of the more resistant LRM would further decrease the tension and move the eye in the opposite direction.

The results of our study indicated that increased relative PDF in the MRM may cause a secondary exaggerated esodeviation when performing contralateral LRM recession as governed by Hering's law. Our results are consistent with Cho et al. [23] finding asymmetric bilateral LRM recession in XT and increased postoperative overcorrection. Greater LRM recession in the deviating eye compared with the fixating eye is associated with greater overcorrection. Cho et al. found that 62% of their patients presented ET overcorrection of >17 PD at day one postoperatively and also suggested that a more-recessed LRM would increase innervation to the contralateral MRM due to Hering's law. Those authors speculated that this mechanism of increased MRM tension resulted in excessive postoperative ET. This hypothesis also corresponded with our earlier study that measured EOM tension in XT [20]. In that study, we found that the PDF of the MRM was larger in the PE than in the NPE in patients with XT. We

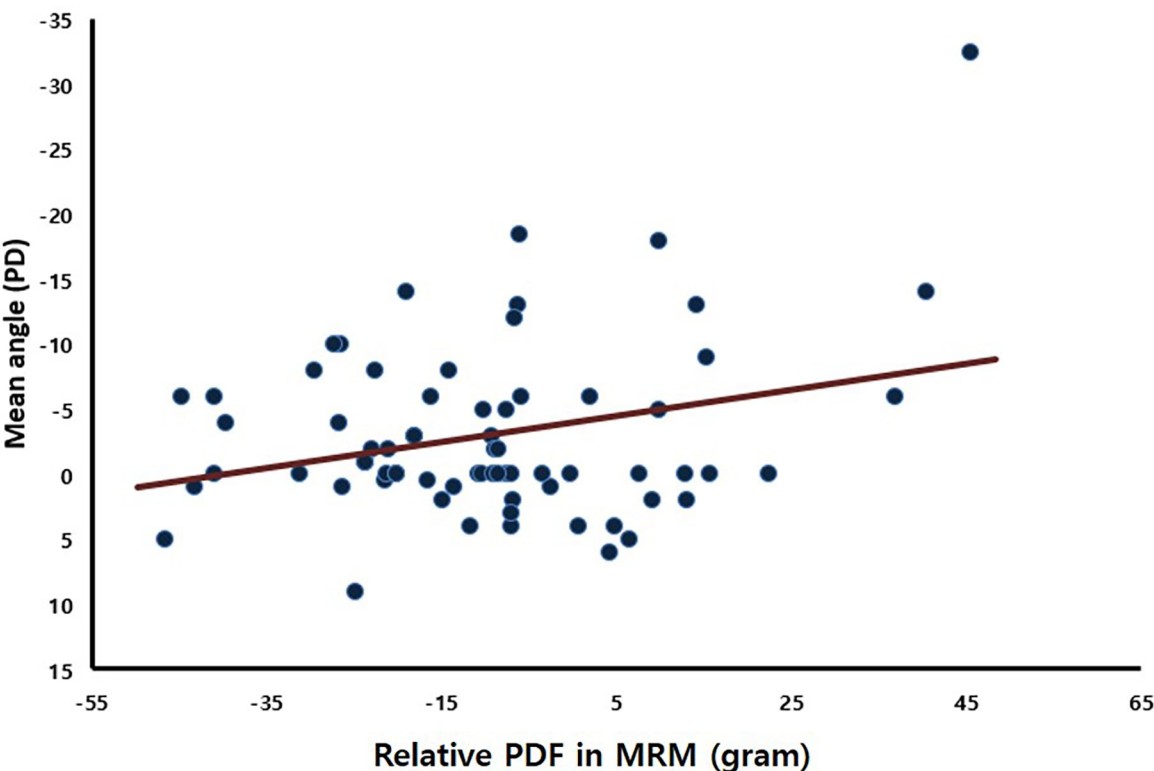

**Fig 4. Correlation between the relative PDF in the MRM in the PE and the average angle of the near and distance deviations (1 month postoperation) in all eyes.** PD, prism diopters (Pearson correlation coefficient = −0.284, p = 0.017). Negative values indicate esodeviation and positive values exodeviation.

believe that the PE will require more tightening of the MRM to maintain its primary position compared to the MRM of the NPE. Consequently, this increased innervational flow to the yoke muscle of the MRM (i.e., the contralateral LRM) might affect the increased PDF of the LRM in the NPE.

The main strength of this study was that our cohort excluded patients who presented possible alternative risk factors for consecutive ET, such as high myopia or hyperopia, amblyopia, lateral incomitance, presence of an "A" or "V" pattern in strabismus, or developmental delay [7, 9, 24]. Comparing the CET and NCET groups revealed no differences in the baseline characteristics including age, duration of manifest deviation, angle of deviation, and stereoacuity. The only significant difference between the two groups was in the relative PDF in the MRM in the PE. Our results provide some evidence for the paradigm that evaluating the PDF in the horizontal rectus muscles (LRM and MRM) would be helpful when considering the plan for strabismus surgery in XT. We believe that a reduction in the surgical amount should be carefully considered in patients with a large PDF in their MRM relative to the ipsilateral LRM to prevent consecutive ET.

Previous evaluations of the PDFs in EOMs in XT surgery suggested an improvement in the predictive accuracy of planned strabismus surgical outcomes. Chronic strabismus may result in fibrosis of the muscle–tendon complex and induce a certain degree of restriction and tension in an EOM that may cause resistance when the eyeball is moving in the opposite direction [14, 15]. Chougule and Kekunnaya reported that in patients with large-angle XT, loss of elasticity and consequently a tight LRM could produce a restrictive leash effect [17]. Previous

studies have suggested that the amount of muscle displacement required for surgery or choosing which eye to operate on should be determined while considering the PDF in the LRM [14, 15]. For example, increased PDF in the LRM may cause resistance when the eyeball is moving in the opposite direction, and decreasing the tension in such cases by recessing the muscle might be beneficial to reduce recurrence and improve the probability of a successful outcome.

In the present study, we chose 10 PD (as used in other studies) as a criterion for consecutive ET [9]. Although some authors have suggested that the amount of overcorrection at 1 day or 2 weeks postoperation can define consecutive ET [8, 23], we defined consecutive ET as this occurring at 1 month postoperation, because a small angle of initial postoperative esodeviation in the early postoperation period after XT surgery often recovers spontaneously to orthophoria within 2–3 weeks. These cases may be observed initially and may not require additional treatment [25].

We recognize the limitations of our study. First, the sample size in the CET group was relatively small due to the strict inclusion and exclusion criteria. Second, all of the cases in our cohort had received recess/resect strabismus surgery. However, Keech and Stewart [26] concluded that the incidence of consecutive ET was not necessarily related to the type of surgery. In this regard, we believe that the results of the present study would be similar to results from bilateral LRM recession. Third, we included patients with basic XT patterns. Prospective studies are therefore still needed to measure the PDF in various types of XT and to clarify whether such tension measurements could help to predict consecutive ET risk.

In conclusion, we identified a possible relationship between the PDF of horizontal rectus muscles and surgical overcorrection in patients with XT. The relative PDF in the MRM can be increased in the PE in patients with consecutive ET after XT surgery. We believe that evaluating the PDF in EOMs provides valuable information about the mechanical properties of these muscles and can improve the accuracy of strabismus surgery, and consequently reduce the risk of consecutive ET. Further studies with larger samples and different surgical techniques may be necessary to confirm the results of this study.

## Supporting information

**S1 Data.**
(XLSX)

## Author Contributions

**Conceptualization:** Hyunkyoo Kang, Hyun Jin Shin.

**Data curation:** Hyunkyoo Kang.

**Formal analysis:** Hyunkyoo Kang.

**Funding acquisition:** Hyunkyoo Kang.

**Investigation:** Hyunkyoo Kang, Hyun Jin Shin, Andrew G. Lee.

**Methodology:** Hyunkyoo Kang.

**Project administration:** Hyun Jin Shin.

**Resources:** Hyun Jin Shin.

**Software:** Hyunkyoo Kang.

**Supervision:** Andrew G. Lee.

**Validation:** Hyunkyoo Kang.

**Writing – original draft:** Hyun Jin Shin.

**Writing – review & editing:** Andrew G. Lee.

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
