## [Decision Letter · Decision Letter 0]

29 Nov 2022

PONE-D-22-23505Risk of Consecutive Esotropia after Surgery for Intermittent Exotropia according to Passive Duction ForcePLOS ONE

Dear Dr. Shin,

Thank you for submitting your manuscript to PLOS ONE. After careful consideration, we feel that it has merit but does not fully meet PLOS ONE’s publication criteria as it currently stands. Therefore, we invite you to submit a revised version of the manuscript that addresses the points raised during the review process.

We look forward to receiving your revised manuscript.

Kind regards,

Abdelrahman M. Elhusseiny

Academic Editor

PLOS ONE

Journal Requirements:

2. Please provide additional details regarding participant consent. In the ethics statement in the Methods and online submission information, please ensure that you have specified what type you obtained (for instance, written or verbal, and if verbal, how it was documented and witnessed). If your study included minors, state whether you obtained consent from parents or guardians. If the need for consent was waived by the ethics committee, please include this information

3. You indicated that you had ethical approval for your study. In your Methods section, please ensure you have also stated whether you obtained consent from parents or guardians of the minors included in the study or whether the research ethics committee or IRB specifically waived the need for their consent.

4. "In your Data Availability statement, you have not specified where the minimal data set underlying the results described in your manuscript can be found. PLOS defines a study's minimal data set as the underlying data used to reach the conclusions drawn in the manuscript and any additional data required to replicate the reported study findings in their entirety. All PLOS journals require that the minimal data set be made fully available. For more information about our data policy, please see http://journals.plos.org/plosone/s/data-availability.

We will update your Data Availability statement to reflect the information you provide in your cover letter."

Reviewers' comments:

Reviewer's Responses to Questions

**Comments to the Author**

1. Is the manuscript technically sound, and do the data support the conclusions?

Reviewer #1: Yes

Reviewer #2: Yes

Reviewer #3: Yes

2. Has the statistical analysis been performed appropriately and rigorously? 

Reviewer #1: Yes

Reviewer #2: Yes

Reviewer #3: Yes

3. Have the authors made all data underlying the findings in their manuscript fully available?

Reviewer #1: Yes

Reviewer #2: Yes

Reviewer #3: Yes

4. Is the manuscript presented in an intelligible fashion and written in standard English?

Reviewer #1: Yes

Reviewer #2: Yes

Reviewer #3: Yes

5. Review Comments to the Author

Reviewer #1: The manuscript is clearly written with reasonable conclusions and discussion of the strengths and weaknesses of the paper. One suggestion to make the discussion more flow more easily is to relate the paragraph starting on line 239 back to the author's own study. The manuscript is sound, however it may be wise to also include how this can be relevant to those without access to the tension measuring device.

Reviewer #2: I want to commend the authors on an interesting paper on a very difficult surgical topic. Over and under correction is always the most challenging and frequent complication of strabismus surgery and they describe an interesting, non-invasive way to try and lower that risk. I think they aptly describe the weaknesses of the study - the small number in the consecutive group especially. They do a very nice job of laying this out though and giving all of the information rather than coming down with concrete conclusions. I think that it would be interesting to see the comparison of this group with bilateral lateral rectus recessions. Overall, this is well written, interesting and formulates reasonable conclusions.

Reviewer #3: 1- Abstract Results: Please mention the number of patients in each group

2- Introduction: The introduction needs to be shortened and focused on the hypothesis of the study. The authors postulate that patients who develop consecutive ET after surgery for XT have either a larger PDF for the medial rectus muscle or smaller PDF of the lateral rectus muscle which explain their exaggerated response to surgery. This should be mentioned clearly in the introduction

3- Line 90: What is the authors’ definition of significant oblique dysfunction?

4- Methods: The surgical dose should be mentioned in the manuscript with references

5- Table 1: What is the unit used for stereopsis?

6- Results: The outcome of the surgeries is not clear. We only know that 8 cases had consecutive ET. What was the mean postoperative angle? Howe many patient was orthotropic? How many had residual XT? Would the authors compare the PDF in those who were orthotropic to those with residual XT looking for higher PDF in the lateral rectus in those with residual XT?

6. PLOS authors have the option to publish the peer review history of their article (what does this mean?). If published, this will include your full peer review and any attached files.

Reviewer #1: No

Reviewer #2: No

Reviewer #3: **Yes: **Ahmed Awadein

---

## [Author Response · Author response to Decision Letter 0]

10 Jan 2023

REQUEST FOR REVISION

PONE-D-22-23505

Risk of Consecutive Esotropia after Surgery for Intermittent Exotropia according to Passive Duction Force

Dear Editorial officer, 

We are deeply grateful for the reviewer’s sincere and valuable comments that have resulted in significant manuscript improvements. We would also like to thank the editorial member of PLOS ONE for their interest in this paper. We hope that our responses satisfactorily address the reviewer’s concerns.

Reviewer #1

The manuscript is clearly written with reasonable conclusions and discussion of the strengths and weaknesses of the paper. 

1) One suggestion to make the discussion more flow more easily is to relate the paragraph starting on line 239 back to the author's own study. 

Thank you very much for your kind review. As your comment, we described the clinical implication of measurement PDF in the last part of the paragraph (line 255-257 of page 13) as following:

“Based on the results of the present study, we believe that a reduction in the surgical amount should be carefully considered in patients with a large PDF in their MRM relative to the ipsilateral LRM to prevent consecutive ET.”

2) The manuscript is sound, however it may be wise to also include how this can be relevant to those without access to the tension measuring device.

Alternative method is forced duction test (FDT). FDT is a simple and easy method for clinically evaluating the mechanical properties of the extraocular muscles (EOMs). However, the results from the FDT are highly dependent on the experience and skill of clinicians, and also this test cannot be used to detect small pathological changes. Thus, the authors have designed a simple and compact device for quantitatively and continuously measuring the passive duction force in EOMs. We hope to soon be able to make our device accessible to other surgeons as well.

Reviewer #2

I want to commend the authors on an interesting paper on a very difficult surgical topic. Over and under correction is always the most challenging and frequent complication of strabismus surgery and they describe an interesting, non-invasive way to try and lower that risk. I think they aptly describe the weaknesses of the study - the small number in the consecutive group especially. They do a very nice job of laying this out though and giving all of the information rather than coming down with concrete conclusions. I think that it would be interesting to see the comparison of this group with bilateral lateral rectus recessions. Overall, this is well written, interesting and formulates reasonable conclusions.

Thank you very much for your kind review and valuable comments. Basically, our surgical management of intermittent exotropia is a R&R procedure. As your suggestion, we will measure the PDF of the lateral rectus in the patient who underwent LROU recession. We believe that the results of this preliminary study would help to plan a further randomized controlled trial (conventional surgical dose group vs. Adjusted surgical dose group according to the results of PDF) in the future. 

Reviewer #3: 

1- Abstract Results: Please mention the number of patients in each group

As your comment, we added the number of patients in each group in the result section of Abstract as following: 

“Of these 70 patients, 8 CET and 62 NCET patients were present.”

2- Introduction: The introduction needs to be shortened and focused on the hypothesis of the study. The authors postulate that patients who develop consecutive ET after surgery for XT have either a larger PDF for the medial rectus muscle or smaller PDF of the lateral rectus muscle which explain their exaggerated response to surgery. This should be mentioned clearly in the introduction

Thank you for your good comment. We changed the introduction section shortened and focused on the hypothesis of the study.

3- Line 90: What is the authors’ definition of significant oblique dysfunction?

‘Significant oblique dysfunction’ indicated inferior (or superior) oblique overaction (or underaction) of +2 or more. We added this point in the Material and Methods section 

4- Methods: The surgical dose should be mentioned in the manuscript with references

As your suggestion, we added the surgical dose as Table 1 (page 6)

5- Table 1: What is the unit used for stereopsis?

Stereoacuity (arcsec) was transformed to log units for analysis. We added this point in the statistical analyses and Table 2 

6- Results: The outcome of the surgeries is not clear. We only know that 8 cases had consecutive ET. What was the mean postoperative angle? Howe many patient was orthotropic? How many had residual XT? Would the authors compare the PDF in those who were orthotropic to those with residual XT looking for higher PDF in the lateral rectus in those with residual XT?

Thank you for your good comment. The mean deviation angles at postoperative 1 month were -3.9±6.5 PD at far and -2.6±7.9 PD at near. 26 patients (37.1%) had ≤10 ET, 19 patients (27.1%) were orthotropic, and 17 patients (24.3%) had residual XT at 1 month after surgery. In response to your comments, we added this in the result section 

As you mentioned, PDF in those with residual XT had a higher relative PDF in the lateral rectus (13.1 vs. 0.1). This is our next research topic “Risk of Undercorrection after Surgery for Intermittent Exotropia according to Passive Duction Force” We will publish this result after this study.

We are deeply grateful for your sincere and valuable comments that have resulted in significant improvements to our manuscript. We have revised our manuscript as suggested and have answered all questions to the best of our abilities. We hope that our responses satisfactorily address your concerns.

---

## [Editor Report · Decision Letter 1]

23 Jan 2023

Risk of Consecutive Esotropia after Surgery for Intermittent Exotropia according to Passive Duction Force

PONE-D-22-23505R1

Dear Dr. Shin,

We’re pleased to inform you that your manuscript has been judged scientifically suitable for publication and will be formally accepted for publication once it meets all outstanding technical requirements.

Kind regards,

Abdelrahman M. Elhusseiny

Academic Editor

PLOS ONE
---

## [Editor Report · Acceptance letter]

8 Feb 2023

PONE-D-22-23505R1 

Risk of Consecutive Esotropia after Surgery for Intermittent Exotropia according to Passive Duction Force 

Dear Dr. Shin:

I'm pleased to inform you that your manuscript has been deemed suitable for publication in PLOS ONE. Congratulations! Your manuscript is now with our production department. 

Kind regards, 

on behalf of

Dr. Abdelrahman M. Elhusseiny 

Academic Editor

PLOS ONE